# Generative complex networks within a dynamic memristor with intrinsic variability

Yunpeng Guo[1,6], Wenrui Duan [2,6] ✉, Xue Liu[3,1,6] ✉, Xinxin Wang[1], Lidan Wang [4], Shukai Duan [4], Cheng Ma [1] ✉ & Huanglong Li [1,5] ✉

Artificial neural networks (ANNs) have gained considerable momentum in the past decade. Although at first the main task of the ANN paradigm was to tune the connection weights in fixed-architecture networks, there has recently been growing interest in evolving network architectures toward the goal of creating artificial general intelligence. Lagging behind this trend, current ANN hardware struggles for a balance between flexibility and efficiency but cannot achieve both. Here, we report on a novel approach for the on-demand generation of complex networks within a single memristor where multiple virtual nodes are created by time multiplexing and the non-trivial topological features, such as small-worldness, are generated by exploiting device dynamics with intrinsic cycle-to-cycle variability. When used for reservoir computing, memristive complex networks can achieve a noticeable increase in memory capacity a and respectable performance boost compared to conventional reservoirs trivially implemented as fully connected networks. This work expands the functionality of memristors for ANN computing.

Connectionism is a movement in cognitive science that hopes to explain mental phenomena using artificial neural networks (ANNs). Since the 1980s, connectionist modelling has gradually gained attention in the field of AI, whose popularity has greatly increased in the past decade due to the success of deep learning (DL). Within DL, researchers study ways of updating the weights of connections to improve the performance of ANNs, starting by defining the architectures of ANNs.

Although DL, as its name suggests, is best known for its multilayer data representation architecture, the invention of new architectural motifs with increasing complexity has enabled DL to continue to make sweeping strides, from AlexNet[1] to ResNet[2], DenseNet[3] and transformer[4]. Along with these advances, interest has quickly turned to architecture design and the possibility of automating architecture engineering towards the more ambitious goal of creating artificial general intelligence[5–7].

Lagging behind the trend of ANNs towards evolvable network architectures, current AI hardware struggles for balance between flexibility and efficiency but cannot achieve both at the same time. GPUs are suitable for general-purpose computing because of their software programmability. However, like other von Neumann processors, GPUs are power-hungry. Rather than being intended for general-purpose computing, ASICs are customized and efficiency-optimized for particular uses, sacrificing post-fabrication software programmability and thus failing to meet the requirement for on-demand ANN architecture generation. This seemingly fundamental conflict between ASIC-like efficiency and software-like programmability will eventually become a road-block for the AI trend towards network architecture evolution.

In contrast to what these familiar computing platforms operate on, the brain principles are completely different, bringing many orders of magnitude higher efficiency than digital methods.

[1]Department of Precision Instrument, Center for Brain Inspired Computing Research, Tsinghua University, Beijing 100084, China. [2]School of Instrument Science and Opto Electronics Engineering, Laboratory of Intelligent Microsystems, Beijing Information Science & Technology University, Beijing 100101, China. [3]School of Integrated Circuits, Tsinghua University, Beijing 100084, China. [4]School of Artificial Intelligence, Southwest University, Chongqing 400715, China. [5]Chinese Institute for Brain Research, Beijing 102206, China. [6]These authors contributed equally: Yunpeng Guo, Wenrui Duan, Xue Liu.
✉e-mail: duanwr10@buaa.edu.cn; liuqingxue@mail.tsinghua.edu.cn; macheng@tsinghua.edu.cn; li_huanglong@mail.tsinghua.edu.cn

The brain has a far more complex network architecture than does any of the existing ANNs. The brain realizes efficient processing of information based on two seemingly opposite principles: segregation and integration. Segregation relies on the spatial aggregation of neurons with similar response preferences to form different functional cortices, while integration relies on communication among the various functional cortices. The structure of the brain network continuously evolves dynamically, disrupting and re-establishing the balance between segregation and integration with sub-second time granularity throughout the lifespan[8]. The brain also uses processes occurring in nature (of course it does) as computational primitives instead of building them up from elementary AND and OR manipulations of 1 and 0, and its components are so highly plastic that they never stop changing in response to the learning environments[9].

To capture this important trait of the brain components, memristive technology is emerging as a promising enabler of the brain-inspired computing paradigm. The memristor, as its name suggests, is a variable resistor with memory. It is most widely used as the emulator of biological synapse and is often integrated into a crossbar array as the neuromorph of the full synaptic connections between two neuron layers in a layered neural network[10]. Heavily influenced by the classical DL practice, these memristive systems have been built primarily as the accelerators for fixed-architecture ANN algorithms[10–17] which in turn demand memristive devices to be static (because typical ANN models are static) and have strictly reproducible behaviors (because typical ANN models are deterministic). To satisfy these demands, however, substantial device-level and circuit-level optimization efforts are required because memristors are, by nature through their internal electrophysical processes, more of dynamic and stochastic devices[18–40] than static and deterministic ones.

In this work, we report a hardware approach that simultaneously exploits the dynamic nature of the memristor and the intrinsic stochasticity in its dynamics to realize the on-demand degeneration of complex networks. With temporal dynamics, Appeltant et al.[41] have proposed the use of a single dynamical node as a complex system by time-multiplexing. In this way, the dynamical node that is reused repeatedly can be treated as a time-domain complex system (i.e., network) composed of a number of virtual nodes with internode couplings (i.e., connections). A number of memristive implementations have also been reported[42], including the use of thin-film oxide memristors[43–48] and memristive nanowire networks[49,50]. We here show that the cycle-to-cycle (C2C) variability of the time constant of the spontaneous resistance decay after the memristor has been electrically excited can be viewed as a source of randomness in connectivity generation, giving rise to nontrivial topological features. In particular, the physically implemented complex networks within a dynamic memristor with intrinsic variability can exhibit a certain degree of small-worldness, lying somewhere between completely regular networks and completely random ones. By regulating the time-slot assignment in multiplexing, networks with different topologies and varying degrees of small-worldness can be generated. Furthermore, we demonstrate the information processing capabilities of several such memristive complex networks folded into the temporal domain in the context of reservoir computing (RC). Experimental results show that the memory capacity of the memristive complex network reservoir is increased to 209.8% of that of the memristive FC network and respectable performance boost in speech recognition tasks compared to conventional reservoirs implemented trivially as fully-connected (FC) networks. The proposed approach of generating complex networks is very generic and applicable to various dynamical memristors with intrinsic variability.

## Results and discussion

### A dynamic memristor with intrinsic variability

The dynamic memristor used in this work has a crosspoint structure vertically stacked as $Pd/HfO_2/Ta_2O_5/Ta$ (50 nm/10 nm/5 nm/20 nm) (see Methods). Its schematic structure and optical spectroscopy image are shown in Fig. 1a, b, respectively. We have also used the focused ion beam (FIB) to prepare the transmission electron microscopy (TEM) specimen. Its cross-sectional TEM image is shown in Fig. 1c, and the corresponding element distribution profiles from energy dispersive spectroscopy (EDS) are shown in Fig. 1d and Supplementary Fig. S1, where individual layers are separable. Figure 1e shows the typical volatile resistance switching characteristics of the memristor. Under a read voltage of 3 V, the device exhibits high resistance about $10^8$ $\Omega$ as estimated from the current through it. When a voltage pulse of the intensity of 5 V and the duration of 1 ms is applied, the current keeps increasing till the pulse is ceased (a read voltage immediately follows). An obvious drop of current from $I_-$ to $I_+$ at the instant the pulse ends can be seen. Over the next few hundred of milliseconds, it is seen that the read-out current $I_+$ gradually decreases until a steady-state value comparable to that measured before pulse application is reached.

In order to understand the nature of the resistance change, electrode area-dependent resistance measurements have been performed. Supplementary Fig. S2 shows the results of DC sweep measurement and the electrical properties of the devices with different areas. The low resistances do not differ significantly from each other, while the high resistance clearly increases with decreasing area, indicating the filamentary nature of the resistance change. This is also consistent with other reported results obtained from devices based on similar materials systems[51].

To evaluate the degree of C2C variation of our device, we perform one thousand identical and independent pulse measurements on this device and analyze its dynamics statistically. As shown in Fig. 1f, the time ($\tau$) of the spontaneous decay of the read-out current $I_+$ varies broadly between 342 ms ($\tau_{min}$) and 1089 ms ($\tau_{max}$). The C2C $\tau$ probability distribution looks like a two-side-truncated Gaussian distribution in which the random variable $\tau$ is bounded both above ($\tau_{max} = 1089$ ms) and below ($\tau_{min} = 342$ ms). It also looks like $\tau$ and $I_+$ are correlated, that is to say, the variation of $\tau$ may originate from the variation of $I_+$, which makes intuitive sense.

A further question then naturally arises: are $I_+$ and therefore $\tau$ also correlated with $I_-$? Behind the question is something important when we consider if the same distribution as acquired from single pulse measurements also reasonably applies to C2C $\tau$ variations measured under arbitrary pulse protocols. It is known that volatile memristors with finite $\tau$s can exhibit paired-pulse facilitation (or short-term facilitation), i.e., $I_-$s increase with each arriving pulse when they are subject to pulse train stimuli as long as pulse intervals are shorter than $\tau$s[19,52]. This can be understood as due to the temporal coupling between the adjacent pulse-induced resistance switching events. Given this, $\tau$s obtained from the last pulses in pulse train or multi-pulse measurements may or may not follow the same truncated Gaussian probability distribution as acquired from single pulse measurements, which is dependent on whether or not $I_+$ and $\tau$ also correlated with $I_-$.

To address this question, we carry out several sets of multi-pulse measurements, each with a different number (2~10) of pulses and containing one thousand independent experiments, and record $I_-$s and $I_+$s at the ends of the last pulses as well as $\tau$s as the last pulses end. The interval between consecutive pulses is set to be 200 ms which is shorter than the minimum recorded $\tau$ in single pulse measurements. This ensures that consecutive resistance-switching events are temporally coupled. As shown in Fig. 1g, h, although the increase in $I_-$ with the number of pulses is statistically significant as the result of the aforementioned paired-pulse facilitation or temporal coupling, $I_+$ and $\tau$ do not have obvious correlations with $I_-$. This observation implies that the

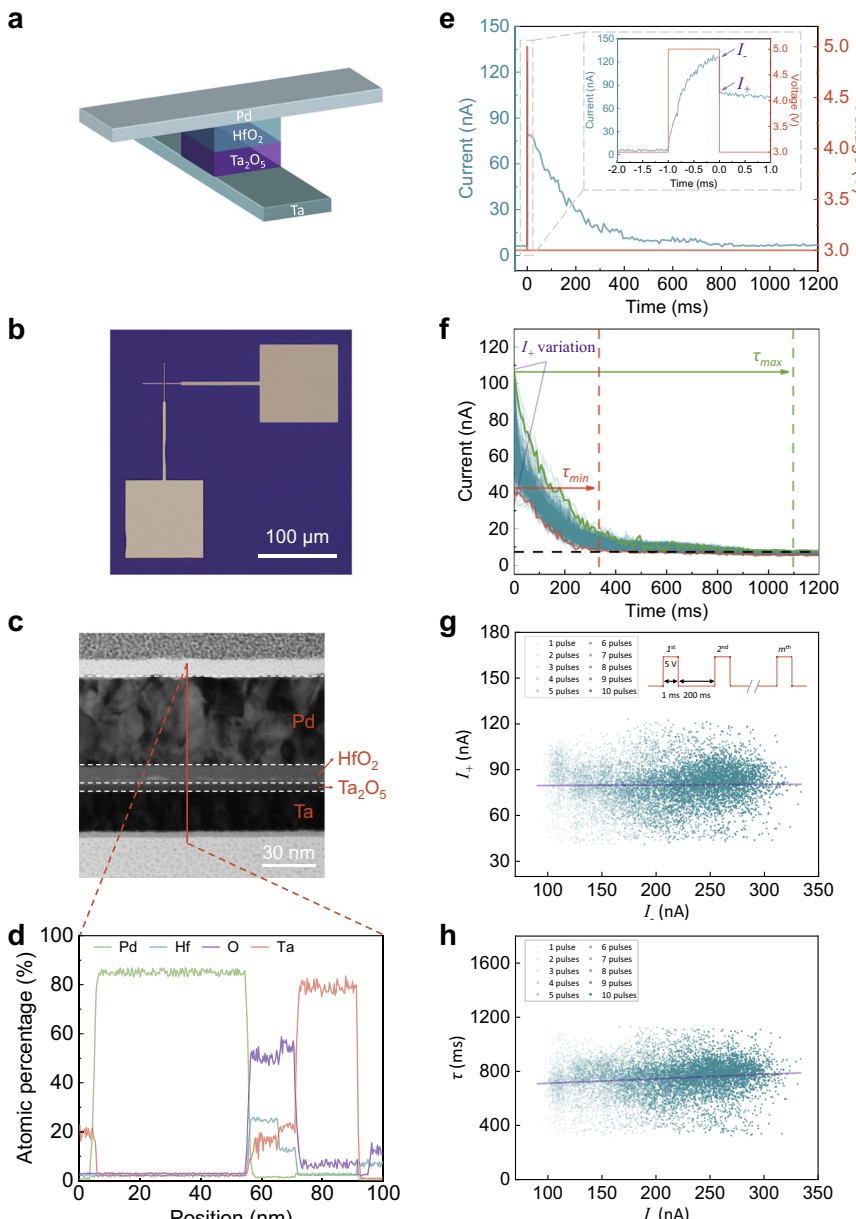

**Fig. 1 | Electrical characteristics of the dynamic memristor. a** Schematic diagram of the structure of the device. **b** Optical microscopy image of a $4 \times 4\ \mu m^2$ device. **c** Cross-sectional TEM image of the dynamic memristor, consisting of a vertically stacked structure of $Pd/HfO_2/Ta_2O_5/Ta$ (50 nm/10 nm/5 nm/20 nm). **d** The corresponding elements distribution profile from EDS. **e** Evolution of the current (cyan curve) through the device under read voltage of (3 V) after the stimulating voltage pulse (5 V and 1 ms) has ceased. Inset: zoom-in view of current evolution over a short time interval before, during and after the stimulating pulse is applied. The

current drops from the peak $I_-$ to $I_+$ immediately after the voltage has decreased from 5 V to 3 V. **f** Spontaneous decay of the current under read voltage from $I_+$ to a baseline steady-state value after the stimulating pulse has ceased. The decay time $\tau$ varies broadly between 342 ms ($\tau_{min}$) and 1089 ms ($\tau_{max}$) over 1000 measurements. **g** Statistical analysis of the correlation between $I_+$ and $I_-$ over 10 sets of multi-pulse (1 - 10) measurements, each containing 1000 independent measurements. **h** Statistical analysis of the correlation between $\tau$ and $I_-$ over 10 sets of multi-pulse (1 - 10) measurements, each containing 1000 independent measurements.

memristive changes in the ionic or electronic configuration of the device induced by multiple pulses are still minor (negligibly affect the $I_+$s measured under 3 V) under our experimental protocols though they are sufficient to be reflected in the $I_-$ instantaneously measured under a relatively large voltage of 5 V. The difference in the sensitivity to the configurational change between $I_-$ and $I_+$ could be due to the strong nonlinearity in the device I-V characteristics; in other words, the memristive changes of the device translate to changes in the instantaneously measured current which increase dramatically with the measuring voltage[24]. Given the noncorrelation between $I_+$ (and $\tau$) and $I_-$, the same C2C $\tau$ probability distribution as acquired from single pulse measurements also reasonably applies to those measured under these

multi-pulse protocols. This is clearly manifested in the well-overlapped distribution functions emerging from the statistical measurements in the respective experimental sets, as shown in Fig. 2a.

As demonstrated previously, the state of a dynamic memristor (like our $Pd/HfO_2/Ta_2O_5/Ta$ memristor) at the present time (or cycle that is discrete and abstracted away from the real continuous physical time) can be temporally coupled to its states at previous times (cycles). In the context of network formation, the temporal coupling between any two cycles is referred to as a connection between two virtual nodes emerging in a sequential fashion in the temporal domain. Therefore, a single memristor can serve as the time-division multiplexed unit that is sequentially reused[41]. The time division multiplexing procedure

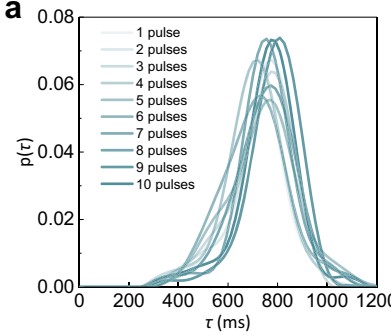

**Fig. 2 | Statistical analysis of the current decay time. a** Probability distributions of $\tau$ obtained in 10 sets of multi-pulse (1–10) measurements, each containing 1000 independent measurements (the same as in Fig. 1h). **b** Fitting of the probability distribution of $\tau$ obtained in single-pulse measurements (cyan curve) by a truncated Gaussian function (purple curve). In between $D_{min}$ and $D_{max}$, the probability

distribution function is a Gaussian function vertically translated by $\varepsilon$ that ensures unity of the probability of the entire sample space. The schematic shows that longer/shorter $\tau$ sampled from the distribution (labeled as a yellow/blue star on the $\tau$ axis) at the moment when a certain virtual node is created enables the formation of connections with subsequent virtual nodes more/less temporally distant away.

reduces the complex network to a single hardware node and therefore facilitates implementations enormously. In addition, the read-out can also be taken at a single point of the delay line. These simplifications will enable ultra-high-speed implementations, using high-speed components that would be too demanding or expensive to be used for many nodes[41,53,54]. As for our dynamical memristor as such a single physical node, it is a passive element with a working current of only a few tens of nA and its speed limit could potentially be in the picosecond range[55], thereby promising speed and energy advantages.

To create a connection between two virtual nodes next to each other, the interval $\theta$ (physical time) between two immediate adjacent cycles must be shorter than $\tau_{min}$; otherwise, these two virtual nodes are temporally independent (supplementary Fig. S3) and are not considered as connected. Therefore, $\theta$ becomes a key tuning factor to modulate network connectivity: given a particular $\tau$, the smaller the $\theta$, the denser the connectivity because the temporal range of coupling ($\tau$) of a virtual node will cover more subsequently emerging ones. What we want to clarify here is that though the weights of the connections are not designed intentionally in this approach, they are naturally present in our physically implemented complex network. Specifically, the connection strength between any two virtual nodes that are temporally separated by m×$\theta$ can be reflected in the amplitude of the remanent current as the result of spontaneous decay over the period of m×$\theta$ from $I_+$ excited at the moment when the former node appears (no further voltage excitation over this period). Accordingly, pairs of virtual nodes with different temporal separations will have different connection strengths. We would also like to remind that virtual nodes appear regardless of whether signals in the form of voltage excitations occur; in other words, the connection strength is pre-defined in principle, though adjustable during the training of the network[56]. Therefore, if voltage excitations do occur during the interval between two nodes, a change in the measured remanent current at the moment when the latter node appears should be regarded as a change in the network state due to the coupling with a different input signal, but not a change in the strength of the connection.

Though networks in the spatial domain can be folded into the temporal domain by multiplexing the dynamic memristor (for given time slots $\theta$s), the generated networks only have trivial topological features if $\tau$ is fixed: the resulting networks are just regular. Quite the reverse, real-world networks are often complex networks that have non-trivial topological features—features that do not occur in simple networks such as regular lattices (e.g., fully connected networks) or totally random graphs. Instead, the structure of a complex network is neither completely regular nor completely random. In this respect, our memristor provides the source of randomness in $\tau$ as described by a

truncated Gaussian C2C probability distribution to guarantee the topological non-triviality of the generated networks.

As shown in Fig. 2b, because $\tau$ varies from C2C that follows a truncated Gaussian distribution, the probability of connection between nodes can be adjusted by $\theta$. Specifically, the probability of connection between a virtual node and a subsequent one temporally separated by physical time less than $\tau_{min}$ is 1; in other words, a node must be connected to $D_{min}$ subsequent nodes, where $D_{min} = \tau_{min}/\theta$. The probability of connection between a virtual node and a subsequent one separated by physical time more than $\tau_{max}$ is 0; in other words, it can by no means be connected to the $D_{max}$th node and beyond after it, where $D_{max} = \tau_{max}/\theta$. As shown in supplementary Fig. S4, the distribution of $\tau$ and therefore the connection probability can be further regulated by pulse amplitude.

## Memristor-inspired 'probabilistic border and all-or-none connection' (PBAONC) complex network model

Basically, there are two approaches to generating a complex network with non-trivial topological features: one is changing the connections between nodes in pristine regularly connected networks[57], and the other is generating connections from scratch[58]. Inspired by the experimentally observed dynamic behavior of our Pd/HfO$_2$/Ta$_2$O$_5$/Ta memristor with intrinsic variability, here we propose a 'probabilistic border and all-or-none connection' (PBAONC) connection generation mechanism for creating complex networks. Starting from an open ring lattice with $N$ nodes, a complex network in which each node forms connections with its clockwise neighbors in an all-or-none (AON) fashion is created under the PBAONC mechanism. To be specific, the clockwise neighbors of a node are classified as either proximal or distal ones according to their distances (measured in the clockwise direction) away from the node under consideration. Each node is connected to all its proximal neighbors but forms no connection with the more distal ones (i.e., AON). For each node, the border between its proximal and distal neighbors is probabilistically determined. Specifically, according to the experimentally characterized distribution of the resistance decay time of the memristor (Fig. 2a), the distance $D_i$ between node $i$ and this border (measured from node $i$ in the clockwise direction) is sampled from the following modeled distribution:

$$
\begin{cases}
p(D) = 0 & \text{if } D < D_{min} \text{ or } D > D_{max} \\
p(D) = \frac{A}{\sigma\sqrt{2\pi}}e^{-\frac{(D-\mu)^2}{2\sigma^2}} + p_0 & \text{if } D_{min} \le D \le D_{max}
\end{cases}
\tag{1}
$$

Where $A = 15.5$, $\mu = \frac{D_{min}+D_{max}}{2}$, $\sigma = \frac{D_{min}+D_{max}}{8.2}$, $p_0 = \frac{1-\int_{D_{min}}^{D_{max}}\frac{A}{\sigma\sqrt{2\pi}}e^{-\frac{(D-\mu)^2}{2\sigma^2}}dD}{D_{max}-D_{min}}$. $D_{min}$ and $D_{max}$ are the two bounds of $D_i$, beyond which the probability

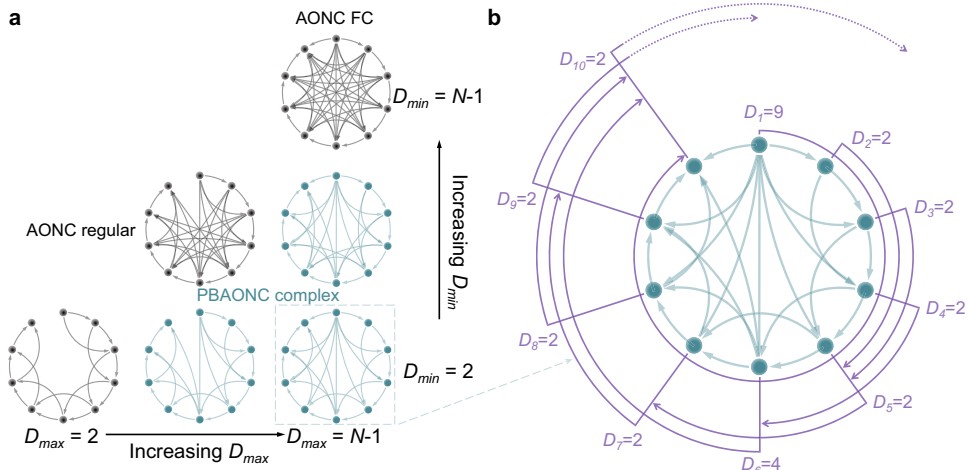

**Fig. 3 | Schematic of the generation of PBAONC complex networks. a** Schematic topologies of the PBAONC networks defined over the parameter space. **b** Schematic of an *N*-node PBAONC complex network (*N* = 10) that are parameterized to $D_{max} = N-1 = 9$ and $D_{min} = 2$. Connections are displayed by thick lines. The connection range of node *i* in the clockwise direction is shown schematically as a thin solid concentric arc starting from the radial line through node *i* and covering the other $D_i$ nodes (The arrows on the lines between nodes indicate the connection direction). For node 9 (10) in this schematic, a part of the (the whole) concentric arc on the clockwise side of node 10 is dashed, indicating that no connection is projected from node 9 (10) to nodes covered by the corresponding dashed arc. For the sake of simplicity, the connection (if present) extended from the first node to the last node (i.e., the *N*th one counted in the clockwise direction) is represented by a short counterclockwise arrow covering the gap between them.

becomes zero. In between $D_{min}$ and $D_{max}$, the probability distribution function is a Gaussian function vertically translated by $p_0$ that ensures unity of the probability of the entire sample space. To some extent, this network generation approach in which connections are sampled from a distance-based probability distribution mimics axonal growth during neuronal development[59]. After the sampling of $D_i$, a specific constraint is imposed that node *N* is the farthest node (measured from node *i* in the clockwise direction) to which node *i* can connect, where *N* is the total number of nodes on the open ring lattice (*i* starts with 1 at the clockwise end of the open ring and increases in the clockwise direction); in other words, if the gap of the open ring lattice is in the clockwise lattice path from *i* to *j*, no connectivity will be projected from node *i* to node *j* even if the distance between them (measured from node *i* in the clockwise direction) is smaller than the sampled $D_i$. The rationale behind imposing this constraint is the law of temporal causality, that is, memristive virtual nodes produced chronologically later should not influence early nodes. As a result, the PBAONC networks are feed-forward or unidirectional. To avoid the appearance of isolated nodes, we also set $D_{min}$ to be nonzero, as illustrated in Fig. 3. Comprehensive analyses of the characteristics of the PBAONC complex networks as compared to the canonical Watts–Strogatz (W-S) small-world (SW) network[57], Erdős–Rényi (E-R) random network[60] and Barabási–Albert (B-A) scale-free network[61] are provided in the Supplementary Figs. S5–S8. Overall, our PBAONC complex networks exhibit a certain degree of small-worldness, achieving functional segregation and aggregation at the same time (see Methods and Supplementary Text).

## Memristive RC using PBAONC complex network reservoirs

As introduced previously, the brain is a powerful computing machine using forbiddingly complex neural networks. One of these connectionist models that exhibits state-of-the-art performance is the RC model[62,63]. A reservoir is a high-dimensional non-linear dynamical system where feed-in inputs are non-linearly transformed into a high-dimensional state space in which different inputs are more easily separable. One of the most prominent advantages of reservoir computing is the simplicity of training that the reservoir itself is left untrained and only the readout layer is required to be trained. Although the exact weight distribution and sparsity are believed to

have limited influence on the reservoir's performance, the best-performing reservoirs have been shown to have spectral radii lower than one[64].

As for memristive reservoirs created through the time-multiplexing procedure[41], Du et al.[43] have used different time-multiplexing time slots for creating different component reservoirs. The motivation was to enrich the reservoir dynamics and benefit from device-to-device variation. Zhong et al.[65] have used a fixed total number of virtual nodes and a fixed time-multiplexing time slot, and investigated the optimal trade-off between the number of component reservoirs and the number of virtual nodes per reservoir. The coupling strength has effectively been tailored in these two cases. A more general framework of network emulation based on a single dynamical system with time-delayed feedback has recently been discussed by several groups[56,66,67]. Among them, Stelzer et al.[56,67] proposed the use of multiple delay loops with different delay lengths for constructing a deep neural network whose interlayer connection topology can be adjusted by the number of delay loops and the delay length of each loop (with a fixed multiplexing time slot and total number of virtual nodes).

Here, we will demonstrate new reservoirs made of our PBAONC complex networks and implemented in dynamic memristors with intrinsic variability. It is clear from the discussions in the last two sections, multiplexing our memristor for *N* cycles with a fixed time slot $\theta$ gives rise to various types of PBAONC networks: if $N \times \theta \leq \tau_{min}$, an FC network (as schematically shown in Supplementary Fig. S9) is created because even the most temporally distant virtual nodes, the first and the last ones, are coupled together; if $\theta \geq \tau_{max}$, then there are only isolated virtual nodes because even the immediately adjacent nodes are uncoupled; if $\tau_{min} < \theta < \tau_{max}$, isolated virtual nodes are still likely to exist. Situations in which $\theta > \tau_{min}$ are beyond our current focus. If $\theta \leq \tau_{min}$ and $N \times \theta \geq \tau_{max}$, each node is coupled to a part of the subsequently emerged nodes that are temporally proximal. With the emergence of a new virtual node in each multiplex cycle, the corresponding temporal border between its proximal and distal neighbor nodes is sampled from the $\tau$ distribution. The workflow of creating the PBANOC complex network physically and the reservoir computing system based on it are schematically shown in Supplementary Fig. S10 and Fig. 4a, respectively.

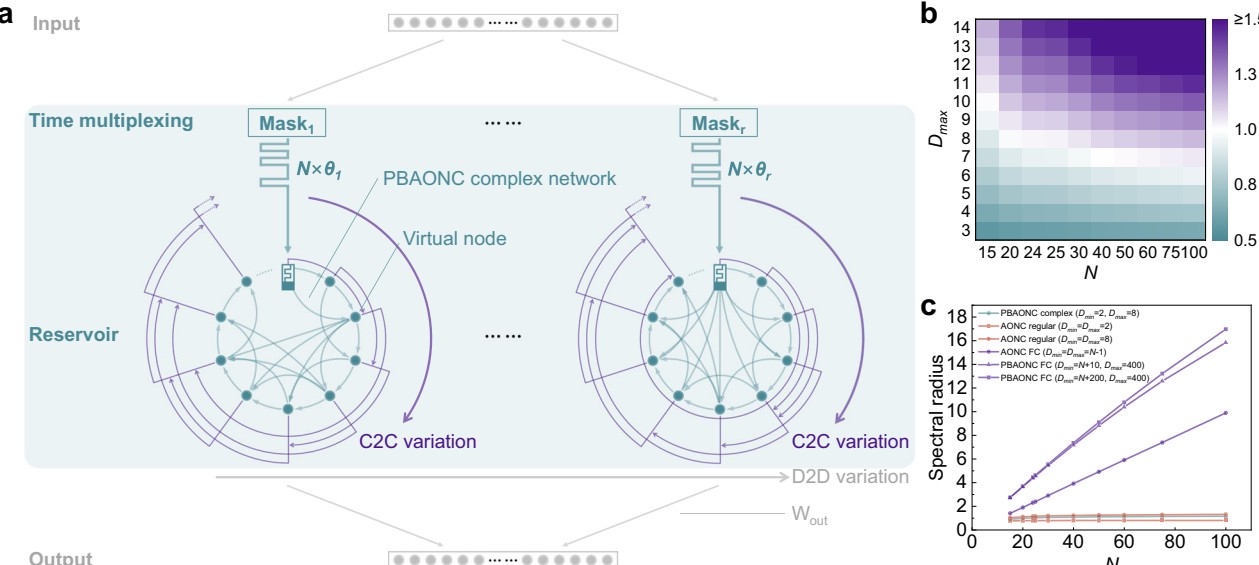

**Fig. 4 | The PBAONC complex network reservoir. a** Schematic of the PBAONC complex network reservoir set for RC based on time multiplexing of the dynamic memristors. For a single component reservoir, a $Q$-dimensional input vector at a certain moment is multiplied with a random $N \times Q$ mask matrix and transformed into an $N$-dimensional vector representing the temporal input stream within the interval $N \times \theta$. This temporal input stream is then fed into the dynamic memristor; in other words, the dynamic memristor is time-division multiplexed with time slot $\theta$ and reused $N$ times. Each component reservoir can have a different time-slot assignment and the transient dynamical responses of each memristor in the same multiplex cycle are aligned with each other in the software. The states of all virtual nodes are linearly weighted through an output weight matrix $W_{out}$ and summed together to obtain the output of the RC system. **b** Contour plot of the spectral radius of the PBAONC complex network's weight matrix as a function of $N$ and $D_{max}$. **c** Spectral radiuses of the weight matrices of the PBAONC complex network ($D_{max} = 8$) and those of the other networks as functions of $N$.

We would like to point out that the weighted summation of the reservoir outputs and the final classification in the testing process, as well as the update of the weight matrix of the output layer in our experimental protocol, are all performed on software. Nevertheless, mixed dynamical and quasi-static memristive reservoir systems have been demonstrated, where quasi-static memristive crossbar arrays are used as the hardware substrate for the readout function[50,68]. The workflows of training and testing our reservoir computing system are shown in Supplementary Fig. S11.

A desired reservoir should exhibit a fading memory, that is, the effect of the previous reservoir state on a future state should vanish gradually as time passes[62]. Practically, this property is assured if the reservoir weight matrix $W$ is scaled so that its spectral radius $\rho(W)$ (i.e., the largest absolute eigenvalue) satisfies $\rho(W) < 1$[64]. Theoretical analyses have also shown that a reservoir has an optimal active state if the $\rho(W)$ is close to 1[69]. Accordingly, in constructing a theoretical model of reservoir, the random weights are routinely drawn from a uniform distribution over $(-\varepsilon, \varepsilon)$ which are then rescaled to a spectral radius less than unity[69,70]. As aforementioned, however, though weights are not designed intentionally in our approach, they are naturally present in our physically implemented complex network. Because each virtual node in our physically implemented PBAONC reservoir is connected to its subsequent ones within its resistance decay time with connection strengths decreasing with temporal separation, we here assign distance-dependent weights to these edges in the simulation. Specifically, the weight is linearly decreased from 0.2 (connection to the immediately following node) as the connected node is farther away. For any node $i$, if $i + D_i \leq N$, the weight of the connection to its border node becomes zero; otherwise, the weight of the connection to node $N$ is $\frac{0.2}{D_i}(D_i + i - N)$. Contour plot (Fig. 4b) shows the $\rho(W)$ of the PBAONC complex network reservoir as a function of $D_{max}$ and $N$. It is seen that as the number of nodes increases the optimal value of $D_{max}$ where the $\rho(W)$ is

closest to 1 reduces, and with $D_{max} = 8$ there are a comparatively wider range of $N$ (20 ~ 30) over which the reservoirs can have their $\rho(W)$s close to 1. By contrast, the $\rho(W)$ of the PBAONC FC network ($N < D_{min}$) reservoir is larger than 1 and increases with the number of nodes (Fig. 4c). This large performance gap (as reflected by the proximity to unity) between the PBAONC complex network and fully connected network under this more physically realistic weight assignment scheme indicates that memristive reservoirs have much room for improvement through the generation of complex networks. The importance of device variability that underpins the generation of complex network topology is also illustrated in Fig. 4c. It can be seen that the trivial AONC regular networks ($D_{min} = D_{max} = 8$ or 2) without randomness in their connectivity patterns have $\rho(W)$s that are less proximal to unity compared to that of the PBAONC complex network, though not as significant as the contrast between the PBAONC complex network and the PBAONC FC network.

Experimentally, different temporal sequences of voltage pulses as inputs to our physically implemented PBAONC complex network reservoir give rise to different trajectories of current evolutions (illustrated in supplementary Fig. S3b, c). The reservoir state is represented by the instantaneous currents obtained when each of the $N$ virtual nodes appears ($I$. if this virtual node is excited by a voltage pulse). These current values are then linearly weighted through an output weight matrix $W_{out}$ and summed together to obtain the output of the reservoir computing system.

Here, we test the time series information processing ability of our PBAONC complex network reservoir in short-term memory (STM) task, parity check (PC) task and spoken-digit recognition task. The STM task is a memory recall task, where the reservoir processes the original time series into a format from which the input values at some time delay in the past can be reconstructed. Details of the STM task implementations are provided in Methods. The memory capacity ($MC_{STM}$) can be quantified by the sum

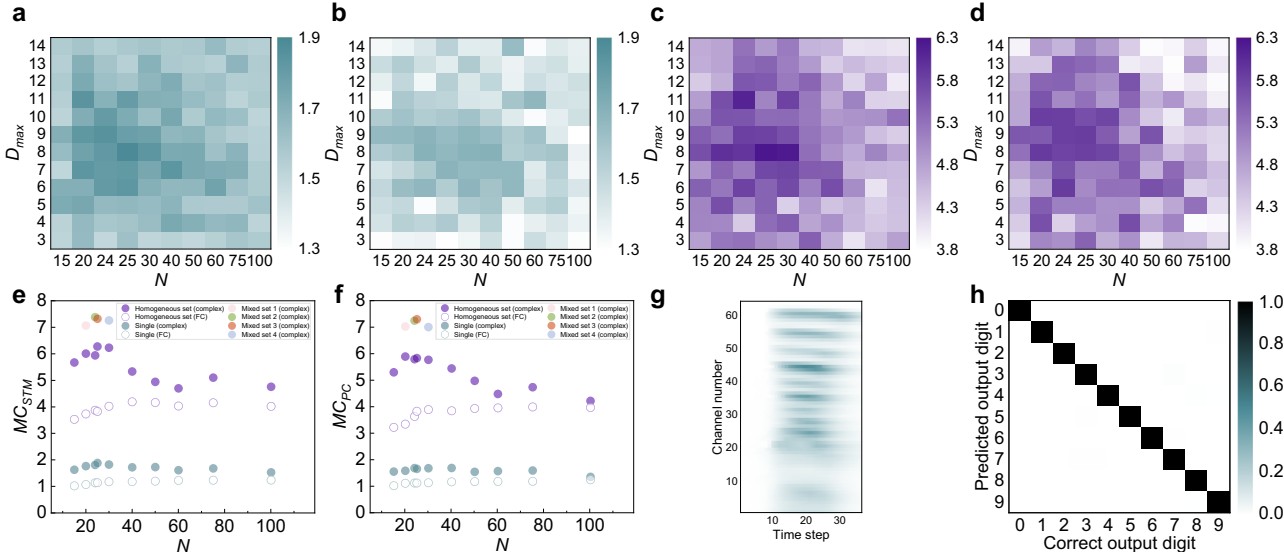

**Fig. 5 | Time-series information processing task implementations.** $MC_{STM}$ (**a**) and $MC_{PC}$ (**b**) of a single PBAONC complex network reservoir implemented with a dynamic memristor. $MC_{STM}$ (**c**) and $MC_{PC}$ (**d**) of a simple multi-device reservoir set (time-slot assignments are the same for each component reservoir) containing 600 virtual nodes in total (each component reservoir contains $N$ virtual nodes). Comparison of $MC_{STM}$ (**e**) and $MC_{PC}$ (**f**) among four mixed reservoir sets, each containing 600 virtual nodes in total but a different number of component reservoirs

(therefore different $N$). Within a set, the time-slot assignments ($\theta$s) for each component reservoir are not all the same (therefore different $D_{max}$). **g** The 64 frequency-channel cochleogram of spoken digit "nine" taken from a female speaker, preprocessed by the Lyon passive ear model (left). **h** Confusion matrix showing the experimentally obtained classification results from the memristive mixed PBAONC complex network reservoir versus the correct outputs.

of the square of the correlation between the output $y_k(t)$ and the delayed input $u(t\text{-}k)$ over all delays as follows:

$$MC_{STM} = \sum_{k=1}^{\infty} Cor^2\big(y_k(t), u(t-k)\big) \tag{2}$$

In addition to the fading memory property, the nonlinear dynamics of the reservoir is also crucial that it allows for linear separability of different inputs, as can be assessed using the PC task. The PC task aims to reconstruct the result of a binary parity operation (e.g., addition operation) over previous inputs up to some delay in the past (e.g., $y_{PC}(m,k) = \sum_{j=0}^{k} u(m-k)(\text{mod}2)$ as the target output). The memory capacity ($MC_{PC}$) is calculated according to Eq. (3):

$$MC_{PC} = \sum_{k=1}^{k=10} Cor^2\big(y_{out}(m,k), y_{PC}(m,k)\big) \tag{3}$$

For the PBAONC complex network reservoirs, contour plots (Fig. 5a, b) show the ratios of $MC_{STM}$ and $MC_{PC}$ to those of the reservoir made of PBAONC FC network, respectively, as functions of the number of virtual nodes ($N$) and $D_{max}$ ($\tau_{max}/\theta$). It is seen that large $MC$s are mainly achieved around $D_{max} = 8$ and $N = 20 \sim 30$, where $\rho(W)$s closest to 1 are achieved according to our weight assignment scheme (Fig. 4b).

To expand the reservoir size or simply generate a set of reservoirs with the same network parameters for each component reservoir (simple reservoir set), multiple devices can be used based on device-to-device (D2D) variations where the reservoir state is represented by the collective states of all devices[43]. The RC performance can be further improved by using different $D_{max}$ parameters for each generated reservoir (mixed reservoir set). Details of the implementations of the simple and mixed reservoir sets are provided in Methods. Figure 5c, d show the contour plots of the ratios of $MC_{STM}$ and $MC_{PC}$ measured for the simple reservoir set to those of the reservoir made of PBAONC FC network, respectively, as functions of the number of virtual nodes ($N$) generated by each single device and $D_{max}$ ($\tau_{max}/\theta$). As expected, multi-device simple reservoir sets have improved $MC$s compared to those of

single-device reservoirs thanks to D2D variation. Four mixed reservoir sets, each with 600 total nodes and containing several best-performing individual PBAONC complex network reservoirs (see Methods), are also investigated. As shown in Fig. 5e, f, these mixed reservoir sets achieve even larger $MC_{STM}$ and $MC_{PC}$, with mixed reservoir set parameterized to have 24 virtual nodes for each of the 25 component reservoirs ($r \times N = 25 \times 24 = 600$) having the largest $MC$, where $r$ is the number of memristors. We use this best-performing mixed reservoir set in the isolated spoken-digit recognition task (see Methods), as shown in Fig. 5g. Figure 5h shows the confusion matrix obtained experimentally during testing. Overall, a recognition rate as high as 99.5% can be achieved in our mixed PBAONC complex network reservoir set. In addition to D2D and C2C variations, this mixed reservoir set further benefits from the richness of temporal dynamics. Nevertheless, our observations (Fig. 5e, f) indicate that respectable performance can already be achieved by simply increasing the number of component reservoirs (still much less hardware overhead compared to that of the conventional parallel feeding procedure) and engineering complex network topology into each individual reservoir (keeping $\theta \le \tau_{min}$ and $N \times \theta \ge \tau_{max}$).

## Discussion

In conclusion, we have demonstrated the potential of simultaneously harnessing both the dynamic nature of the emerging memristor device and the intrinsic stochasticity in its dynamics for the on-demand generation of our co-designed PBAONC complex networks with desired topological features, echoing an emerging trend in the field of connectionist AI towards evolving the architectures or topologies of neural networks (architecture engineering). In this memristive implementation approach, the entire topological complexity of the PBAONC complex networks can be folded into the temporal domain by reusing the memristor device repeatedly in a time-division multiplexed manner, and the network connectivity is developed with the emergence of new virtual nodes over time as a temporal unfolding of the memristor's dynamics. Though perfect homogeneity, in addition to mitigated hardware overhead, has been viewed as one of the main advantages of

using a single dynamical node as a complex system[56], our approach actually benefits from exploiting the intrinsic C2C variability of the memristor's resistance decay dynamics in generating non-trivial network connectivity patterns. In particular, the generated PBAONC complex networks exhibit a certain degree of small-worldness, a feature that is ubiquitous across biological (e.g., the brain), technological, and social networks, and accounts for the optimal balance of functional segregation and integration in the brain network. Finally, we have illustrated the advantages of our memristive PBAONC complex networks in the brain-inspired RC tasks. Experimental results show that the *MC* of the memristive complex network reservoir is increased to 209.8% of that of the memristive FC network and respectable performance boost in speech recognition tasks compared to conventional reservoirs implemented trivially as FC networks, which may be accounted for by their nontrivial topological features (e.g., a certain degree of small-worldness and close-to-one $\rho(W)$). This work may represent a paradigm shift in neuromorphic computing or machine learning with memristors and serves as a springboard for more studies and applications of the intrinsic physical nature of memristors, such as dynamics and stochasticity, for new computing architectures.

## Methods

### Device fabrication

The dynamic memristor was fabricated into a $2 \times 2\,\mu m^2$ cross-point structure on a silicon substrate with 300 nm thermally grown silicon oxide on it. 20-nm Ta was deposited first on the substrate by radio frequency (RF) sputtering and patterned by photolithography as the bottom electrode. Photolithographically patterned $Ta_2O_5$ (5 nm) and $HfO_2$ (10 nm) were then deposited by RF sputtering. Finally, the 50-nm top Pd electrode was deposited and lithographically patterned.

### Electrical measurement

Cyclic quasi-DC voltage sweep measurements were performed by the Keysight B1500A semiconductor analysis system. The Keysight B1530A waveform generator/fast measurement unit was used to perform the pulse measurements. Using a two-probe (W tips) configuration, DC and pulsed voltages were applied to one electrode with the other electrode grounded.

For the STM and PC tasks, we use a binary time series input with a stochastic "0" or "1" component in each time step. In any time step, the corresponding series component is multiplied with a randomly generated (fixed throughout the processing task) binary mask matrix (functionally equivalent to a synaptic weight matrix) of the size of $N \times 1$, where $N$ is the number of nodes in the reservoir, thereby producing a new $N$-dimensional vector. By time-division multiplexing, each virtual node is updated using the corresponding vector component of the $N$-dimensional vector. At the end of each time step, all virtual nodes have been updated and the reservoir reaches a new state, ready to process input in the next time step. Experimentally, the "0" and "1" vector components of the *N-dimensional* vector signal are represented by 1-ms voltage pulses of the intensities of 3 V and 5 V, respectively, with an interval $\theta$ between successive pulses ($\theta$ is also referred to as the multiplex cycle duration). As such, a time series component is held for a duration of $N \times \theta$ after which the component in the next time step will be processed by the reservoir. As discussed in the main text, various types of PBAONC network reservoirs can be generated by multiplexing a single dynamic memristor, depending on the number of multiplexing cycles $N$ (i.e., the number of virtual nodes) as well as the multiplex cycle duration $\theta$. The reservoir's transient dynamical response is read out by an output layer (implemented in software), which are linear weighted sums of the reservoir node states (i.e., I-s). Note that a major advantage of RC is fast training because only weights in the linear readout layer need to be trained, while the connections in the reservoir remain fixed. The training is also performed using software.

**Table 1 | Mixed reservoir set**

| Reservoir set | Mixed set 1 | Mixed set 2 | Mixed set 3 | Mixed set 4 |
|---|---|---|---|---|
| $D_{max}$ | Amount | Amount | Amount | Amount |
| 3 | 1 | 1 | 1 | 1 |
| 4 | 1 | 1 | 1 | 1 |
| 5 | 2 | 1 | 1 | 1 |
| 6 | 5 | 4 | 4 | 3 |
| 7 | 5 | 4 | 4 | 3 |
| 8 | 5 | 4 | 4 | 3 |
| 9 | 5 | 4 | 4 | 3 |
| 10 | 2 | 2 | 1 | 1 |
| 11 | 1 | 1 | 1 | 1 |
| 12 | 1 | 1 | 1 | 1 |
| 13 | 1 | 1 | 1 | 1 |
| 14 | 1 | 1 | 1 | 1 |
| $r$ | 30 | 25 | 24 | 20 |

For the isolated spoken-digit recognition task, the inputs for the reservoir are 64-frequency channel sound waveforms of isolated spoken digits (0–9 in English) from the NIST TI46 database. 450 out of 500 audio samples in the TI-46 database are selected for training, and the remaining 50 samples are used for testing. We use 25 devices to implement the RC system. Input signal through each independent channel is binarized to a 36-time step 0/1 time series. The series component in each time step is multiplied by a randomly generated binary mask matrix of the size of $24 \times 1$, which is represented by a train of 3 V or 5 V pulses (1 ms in duration). Though $\theta$s (or pulse intervals) for each device in the mixed reservoir set can be different, their transient dynamical responses in the same multiplex cycle are aligned with each other in the software for further processing.

### Complex network performance indicators

The calculations of all network topology indicators were performed by the Python library NetworkX.

### Mixed reservoir set

The total number of virtual nodes for each mixed reservoir set is 600. It can be seen from Fig. 5a–d in the main text that high-quality reservoirs can be found mainly at $D_{max} \in \{6, 7, 8, 9\}$ in the parameter space. Therefore, our approach to constructing a good mixed reservoir set is using many reservoirs with $D_{max} \in \{6, 7, 8, 9\}$ and supplementing with other reservoirs with $D_{max} \in \{3, 4, 5, 10, 11, 12, 13, 14\}$ to benefit from the richness of temporal dynamics. The parameters defining the four mixed reservoir sets tested in this work are shown in Table 1.

## Data availability

All data needed to evaluate the conclusions in the paper are present in the paper and/or the Supplementary Materials. Additional data related to this paper is available from the authors upon reasonable request. Source data are provided with this paper.

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

## Acknowledgements

The authors acknowledge funding from National Natural Science Foundation (grant nos. 61974082, 61704096, 61836004), National Key R&D Program of China (2021ZD0200300, 2018YFE0200200), Youth Elite Scientist Sponsorship (YESS) Program of China Association for Science and Technology (CAST) (no. 2019QNRC001), Key Laboratory of Luminescence Analysis and Molecular Sensing (Southwest University), Ministry of Education, Southwest University, Chongqing, 400715, PR China, Tsinghua-IDG/McGovern Brain-X program, Beijing science and technology program (grant nos. Z181100001518006 and Z191100007519009), Suzhou-Tsinghua innovation leading program 2016SZ0102, and CETC Haikang Group-Brain Inspired Computing Joint Research Center.

## Author contributions

H.L. and Y.G. convinced the idea. H.L. supervised the project. Y.G. and X.W. fabricated the devices. Y.G. and W.D. performed the simulations. L.W. and S.D. assisted with the simulations. Y.G., X.L., and C.M. performed device characterizations. Y.G. and H.L. wrote the manuscript with input from all authors.

## Competing interests

The authors declare no competing interests.
