## [Peer Review File · Nature Communications]

REVIEWER COMMENTS

Reviewer #1 (Remarks to the Author):

I'm pleased with the manuscript by Yunpeng Guo and Huanglong Li. The main idea here is to project temporal dynamics of a memristor to spatial/physical connectivity patterns in bio-inspired networks. The broad idea is not new, but the specific idea and its implementation in the paper are certainly new - something I had not thought of. I am an expert in memristive devices and neural networks in general, so I find most of the claims to be legitimate. However, I am not an expert in all the math behind reservoir computing, so (to the editor) please make sure you run those by another reviewer. Some comments to improve the scope of the manuscript:

1. There is very little detail on the device itself. There are some IV curves in the supplement, but I would have liked to see more of the devices details (e.g., schematic, pictures) included in the main text (Fig. 1). This is partly important because the main results of the paper are based on single memristors. However, I would also like the authors to acknowledge that similar results can be obtained using a variety of memristive dynamics.
2. Please illustrate the pulsing scheme used in Fig. 1c - it's not apparent by looking at the figure. Also, please illustrate the sentence in a figure panel: "" In between D_{min} and D_{max} , the probability distribution function is a Gaussian function vertically translated by ϵ that ensures unity of the probability of the entire sample space."
3. The English language in the paper (and the supplement) must be improved before publication, though it is readable for now.
4. The abstract is very difficult to understand. I would eliminate most of the jargon (including connectionism) and simply say that you are exploiting device dynamics to determine/generate physical connectivities in an AI network. And that you're demonstrating its utility in an RC, where dynamical connectivity is important.
5. I suggest not using the phrase "complex" to refer to high-dimensional spaces. Instead, simply call it high-dimensional connectivity (or representation) and "dynamical tuning" of network structure. The word 'complex' can distract the readers from the main message.

6. I would like to see more schematics or flow charts of how you set up the read outs from a memristor and translated them to connectivity patterns. The math is good, but the experimental workflow will help, especially in the supplement. The "electrical methods" section is not detailed enough and lacks illustrations.

Finally, please acknowledge that a full hardware implementation of mapping the dynamical temporal response of a memristor to physical and spatial connections is unclear and not demonstrated at large scales, especially on fully integrated circuits.

Reviewer #2 (Remarks to the Author):

Summary

In this manuscript, Guo et al. report an approach to generate probabilistic network models based on time-multiplexing of the dynamic memristor. The authors claim that the decay time of dynamic memristor has a truncated Gaussian distribution. And by applying pulse stimulations with certain time intervals, each stimulation is treated as a virtual node of the network, and one virtual node is considered to only connect to the following virtual nodes lying in its decay time. The stochasticity of the delay time enables dynamic memristors to generate complex network reservoirs in this way and to achieve good performance in reservoir computing tasks. The manuscript is well organized and developed. However, there are several evident issues within the manuscript regarding device dynamics analysis, experiment description and the prospects of the approach, which are listed below.

Issues

- The authors assume that a virtual node (V_1) is connected to all the following virtual nodes (V_2, V_3, \dots) that are within its decay time. This raises questions on two aspects. On the one hand, the authors treat all the virtual nodes within the delay time (V_2, V_3, \dots) the same. However, the virtual node appearing at different time points within the delay time may have different connection strength with V_1 . Besides, the virtual nodes appearing in the front (e.g., V_2) may influence the connection strength of the latter virtual nodes (e.g., V_3) with V_1 . On the other hand, there is no evidence showing that a virtual node is totally independent from the virtual nodes outside of its decay time. They may still have relations and need to be considered as connected. Thus, this assumption is not convincing and requires many further analyses to confirm.

- The authors assign a random weight value between -0.5 and +0.5 to each connection. This is very controversial from the first intuition that connection weight decreases if the virtual nodes are further from each other. The authors need to prove if their method is reasonable.
- The authors do not describe clearly what are the transient dynamical responses of the memristor that are used as input to the following linearly weighted matrix. Furthermore, the authors do not show how the transient dynamical responses are correlated to the memristor inputs. There need to be some experiment demonstrations showing the inputs and outputs of the generated network.
- The authors do not clearly describe how and when the learning and testing processes are done. It would be great if workflow schematics can be used to describe the details, such as when the training data is input, when the weight matrix is updated, when the training is done and when the testing is performed.
- According to the authors' description, the transient dynamic responses of all the virtual nodes need to be recorded and to be further aligned. This could be very power consuming and time consuming. Besides, each memristor needs to be optimized separately to work with their optimum time slot, which is not promising for large-scale applications and is also very inefficient. These are some intrinsic and crucial drawbacks of the approach in the manuscript.

Reviewer #3 (Remarks to the Author):

The manuscript presents an experimental implementation of a reservoir computer based on a single memristor and a time-multiplexing technique. Due to the time-multiplexing, an equivalent virtual network has a feed-forward structure, where the coupling between the virtual nodes occurs due to the finite current decay time in the device.

The main idea of this work is based on the intrinsic cycle-to-cycle variability. As a result of such a variability, the spontaneous current decay time is not constant and has a stochastic nature. Therefore, the resulting virtual network becomes non-regular. In addition, the virtual network can be controlled by varying the time-multiplexing stepsize. For sufficiently small stepsize, all virtual nodes are coupled, resulting in a Fully Coupled (FC) network. The time-multiplexing stepsize is the metaparameter that is used in the paper to switch between FC, small-world, or uncoupled networks.

Apart from the experimental implementation, the authors compare the performance of the obtained reservoir with the extreme FC case. This comparison shows that the performance increases significantly when the network topology becomes non-trivial (non-FC).

I find the idea of exploiting the intrinsic variability of the experimental device for reservoir computation very interesting. The intrinsic variability increases the complexity of the system and can potentially lead to improvement.

On the other hand, in my opinion, the paper does not show that the intrinsic variability is the source of the improvement. Perhaps I have misunderstood, but their only message regarding the performance is that their setup in the properly tuned regime with non-trivial network topology can perform better than the same setup in the FC regime. Whether the improvement is due to network variability or the fact that it is not FC is unclear.

Some other comments are listed below.

In my opinion, the paper is not suitable for publication in Nature Communications in the present form. A major revision may make it suitable, depending on the ability of the authors to refute the criticism.

Comments:

- With reference to my comment above, can you show that the improvement in reservoir performance is related to the network variability?

- Although time-multiplexing and time-folded virtual network is one of the main points used in the manuscript, previous main results on this topic are not properly introduced.

- The manuscript "promises" a scheme with on-demand generation of the network. However, the presented scheme is essentially based on the variation of the time-multiplexing stepsize " θ " to achieve different coupling topologies. I am afraid that this possibility has already been explained in several previous studies. The authors did not mention this fact in the introduction.

- It is unclear where the ring topology for the visual representations of the virtual network in Figures 3,4 comes from. Is the last node N connected to the first node 1 via a delayed feedback? At least I did not find this in the description.

- When the spectral radius of the reservoir is computed, random weights from the interval $[-0.5, +0.5]$ are assigned. It is unclear to me how these random weights relate to the experimental reservoir used for the computations. Are the weights between the virtual nodes additionally weighted? (I guess not). If these links are not weights, why are additional weights used for the spectral radius (especially negative values)?

- The authors should clearly state that their virtual network is feed-forward, i.e., it is a directed network, where the earlier in time virtual nodes influence the later virtual nodes. In the literature on complex networks, FC networks are mostly considered to be bidirectional. Perhaps the directionality can also be shown in the figures. What is the coupling weight matrix for the typical (or best) choice of parameters?

Point-by-point response to reviewers' comments

Reviewer #1

General comments:

I'm pleased with the manuscript by Yunpeng Guo and Huanglong Li. The main idea here is to project temporal dynamics of a memristor to spatial/physical connectivity patterns in bio-inspired networks. The broad idea is not new, but the specific idea and its implementation in the paper are certainly new - something I had not thought of. I am an expert in memristive devices and neural networks in general, so I find most of the claims to be legitimate. However, I am not an expert in all the math behind reservoir computing, so (to the editor) please make sure you run those by another reviewer. Some comments to improve the scope of the manuscript.

Response:

Thank you very much for recognizing the originality of our work and giving us insightful comments to further improve the quality of this work. Based on your comments, we have performed more thorough testing and characterization of the device, and optimized the texts and diagrams of the manuscript to improve the readability.

Our responses to your specific comments one by one are shown as follows.

Comment #1:

There is very little detail on the device itself. There are some IV curves in the supplement, but I would have liked to see more of the devices details (e.g., schematic, pictures) included in the main text (Fig. 1). This is partly important because the main results of the paper are based on single memristors. However, I would also like the authors to acknowledge that similar results can be obtained using a variety of memristive dynamics.

Response:

We would like to thank the reviewer for pointing out this issue, which is important for the readers to better understand the structural and electrical properties of our Pd/HfO₂/Ta₂O₅/Ta (50 nm/10 nm/5 nm/20 nm) dynamic memristor, as well as the mechanism of resistance change. In the revised manuscript, we have provided its optical microscopy image and supplemented with a schematic diagram of the three-dimensional structure of the device in order to more straightforwardly reflect the device structure, as shown in Figs. 1a,b. To further characterize the pristine device, we have used focused ion beam (FIB) to prepare the transmission electron microscopy (TEM) specimen. Its cross-sectional TEM image is shown in Fig. 1c, and the corresponding element distribution profiles from energy dispersive spectroscopy (EDS) are shown in

Fig. 1d and Fig. S1, where individual layers are separable. The added text is blue marked on page 4 of the revised manuscript.

To understand the nature of the resistance change, electrode area dependent resistance measurements have been performed. Fig. S2e shows the electrical properties of the devices with different areas obtained under voltage sweeping. The low resistances do not differ significantly from each other, while the high resistance clearly increases with decreasing area, indicating filamentary nature of the resistance change. This is also consistent with other reported results obtained from devices based on similar materials systems¹. The added text is blue marked on page 5 of the revised manuscript.

As you have pointed out, this proposed approach of generating complex networks is applicable to various dynamical memristors with intrinsic variability. In fact, dynamics and its variability are intrinsic properties of memristors that are gaining increasing attention in recent years as important sources of computational power²⁻⁵. This statement has been added to the revised manuscript on page 4.

Comment #2:

Please illustrate the pulsing scheme used in Fig. 1c - it's not apparent by looking at the figure. Also, please illustrate the sentence in a figure panel: "" In between D_{min} and D_{max} , the probability distribution function is a Gaussian function vertically translated by ϵ that ensures unity of the probability of the entire sample space."

Response:

Thank you very much for these suggestions. In the revised manuscript, we have shown the pulse schemes as an inset in Fig. 1g (originally Fig. 1c). As for the illustration of the probability distribution function, we have modified the legend label of the model distribution function that is fitted to the experimental data in Fig. 2b to make the illustration clearer.

Comment #3:

The English language in the paper (and the supplement) must be improved before publication, though it is readable for now.

Response:

We sincerely appreciate the reviewer's comment. We have carefully polished our manuscript and enhanced its readability.

Comment #4:

The abstract is very difficult to understand. I would eliminate most of the jargon (including connectionism) and simply say that you are exploiting device dynamics to determine/generate physical connectivities in an AI network. And that you're demonstrating its utility in an RC, where dynamical connectivity is important.

Response:

We sincerely appreciate the reviewer's comment. We have crafted the abstract to make ourselves understood more easily.

Comment #5:

I suggest not using the phrase "complex" to refer to high-dimensional spaces. Instead, simply call it high-dimensional connectivity (or representation) and "dynamical tuning" of network structure. The word 'complex' can distract the readers from the main message.

Response:

Thank you very much for your comment. Here, “complex network” is an umbrella term used in the field of network science. By definition, a complex network is a graph (network) with non-trivial topological features—features that do not occur in simple networks such as regular lattices (e.g., fully connected networks) or totally random graphs. Instead, the structure of a complex network is neither completely regular nor completely random. One of the most famous types of complex networks is the small-world network⁶, characterized by short characteristic path length and high clustering coefficient.

By exploiting device dynamics with intrinsic cycle-to-cycle variability, our approach can generate networks with different degrees of small-worldness as required within a single memristor. This has not been realized in previous studies where only regular networks (more specifically, fully connected networks) were physically implemented.

We have defined the terminology “complex network” in the revised manuscript on page 9.

Comment #6:

I would like to see more schematics or flow charts of how you set up the read outs from a memristor and translated them to connectivity patterns. The math is good, but the experimental workflow will help, especially in the supplement. The "electrical methods" section is not detailed enough and lacks illustrations.

Finally, please acknowledge that a full hardware implementation of mapping the dynamical temporal response of a memristor to physical and spatial connections is unclear and not demonstrated at large scales, especially on fully integrated circuits.

Response:

Thank you very much for your suggestions. In the revised manuscript, we have provided a workflow chart (Fig. S10) of how we set up the readouts from the memristor and translate them to connectivity patterns.

We have also detailed the experimental protocols for the STM task, PC task and isolated spoken-digit recognition task, respectively, in the “Electrical measurement” section of the revised manuscript.

As for your comment on the full hardware implementation of mapping of the physical and spatial connections to the dynamical temporal response of a memristor, the time division multiplexing procedure offers considerable practical advantages that the reduction of a complex network to a single hardware node facilitates implementations enormously, because only a few components are needed, which may in turn reduce energy consumption significantly. In addition, the read-out can also be taken at a single point of the delay line. These simplifications will enable ultra-high-speed implementations, using high-speed components that would be too demanding or expensive to be used for many nodes⁷⁻⁹. As for our dynamical memristor as such a single physical node, it is a passive element with working current of only a few tens of nA and its speed limit could potentially be in the picosecond range¹⁰, thereby promising speed and energy advantages.

In this work, we have applied our physically implemented PBAONC complex network to reservoir computing. Note that a major advantage of reservoir computing is fast training because only weights in the linear readout layer need to be trained, while the connection weights (not intentionally pre-designed but naturally present in our physical reservoir) in the reservoir remain fixed. In our experimental protocol, the weighted summation of the reservoir outputs and the final classification in the testing process, as well as the update of the weight matrix of the output layer are all performed on software. Nevertheless, mixed dynamical and quasi-static memristive reservoir systems have been demonstrated, where quasi-static memristive crossbar arrays are used as the hardware substrate for the readout function^{11,12}.

We have added these discussions in the revised manuscript on pages 8 and 12.

Reviewer #2

General comments:

In this manuscript, Guo et al. report an approach to generate probabilistic network models based on time-multiplexing of the dynamic memristor. The authors claim that the decay time of dynamic memristor has a truncated Gaussian distribution. And by applying pulse stimulations with certain time intervals, each stimulation is treated as a virtual node of the network, and one virtual node is considered to only connect to the following virtual nodes lying in its decay time. The stochasticity of the delay time enables dynamic memristors to generate complex network reservoirs in this way and to achieve good performance in reservoir computing tasks. The manuscript is well organized and developed. However, there are several evident issues within the manuscript regarding device dynamics analysis, experiment description and the prospects of the approach, which are listed below.

Response:

We are glad that this manuscript has left positive impression on the reviewer and we thank you very much for your valuable input to help improve our manuscript. According to your comments, more comprehensive experimental studies and more detailed analyses have been conducted.

Our responses to your specific comments one by one are shown as follows.

Comment #1:

The authors assume that a virtual node ($V1$) is connected to all the following virtual nodes ($V2, V3, \dots$) that are within its decay time. This raises questions on two aspects. On the one hand, the authors treat all the virtual nodes within the delay time ($V2, V3, \dots$) the same. However, the virtual node appearing at different time points within the delay time may have different connection strength with $V1$. Besides, the virtual nodes appearing in the front (e.g., $V2$) may influence the connection strength of the latter virtual nodes (e.g., $V3$) with $V1$. On the other hand, there is no evidence showing that a virtual node is totally independent from the virtual nodes outside of its decay time. They may still have relations and need to be considered as connected. Thus, this assumption is not convincing and requires many further analyses to confirm.

Response:

Thank you very much for your comment. Our approach can create nontrivial network topologies as desired via controlling the multiplexing time slot θ , thanks to the variability of the resistance decay time τ (obeying a certain distribution) of the dynamical memristor. This approach is premised on the basis that the edge (unweighted) between any two virtual nodes is formed if they are dynamically coupled, or in other words, their temporal separation is less than τ . On the other hand, edges in a network are often associated with weights that differentiate them in terms of their strength. What we want to clarify here is that though weights are not designed

intentionally in our approach, they are naturally present in our physically implemented complex network. Specifically, the connection strength between any two virtual nodes that are temporally separated by $m \times \theta$ can be reflected in the amplitude of the remanent current as the result of spontaneous decay over the period of $m \times \theta$ from I_+ excited at the moment when the former node appears (no further voltage excitation over this period). Accordingly, pairs of virtual nodes with different temporal separations will have different connection strength, just as the reviewer has pointed out.

We have added these discussions in the revised manuscript on page 8.

As for the reviewer's comment on whether the strength of the connection between any two virtual nodes is predefined or not, we would like to remind that virtual nodes appear regardless of whether signals in the form of voltage excitations occur; in other words, the connection strength is pre-defined in principle, though adjustable during the training of the network¹³. As above-mentioned, for any two virtual nodes, if there is no further voltage excitation over their time interval (voltage excitation only occurs at the same time as the former node appears), the measured remanent current at the moment when the latter node appears is a reflection of the strength of connection between this pair of nodes. If voltage excitations do occur during the interval, this measured current may be affected. However, this should be regarded as a change in the network state due to the coupling with a different input signal, but not a change in the strength of connection.

We have added these discussions in the revised manuscript on page 8.

Finally, with regard to the reviewer's comment on whether two virtual nodes with temporal separation greater than the resistance decay time τ are coupled or not, we want to clarify that in our work coupling between any two nodes are indicated by the existence of dependency of the voltage excited state (measured by I) of the latter node on whether the former node has been excited or not (no further voltage excitation over their interval). We have explicitly shown in Fig. S3a of the revised manuscript that I upon excitation of a node is negligibly influenced by the immediately preceding node if their temporal separation is greater than τ_{max} (the maximum achievable value of τ) even if the former one has been excited. We totally agree with the reviewer that two nodes separated by interval greater than τ could somehow still be correlated. This correlation may be manifested by the dependency of some other state variables of the latter node on those of the former one over a longer time scale. Based on the results shown in Figs. 1g,h, however, these state variables, if present, will not affect the evolution of the resistance state of the memristor, leaving any two nodes that are separated by θ greater than τ uncoupled according to our definition.

Comment #2:

The authors assign a random weight value between -0.5 and +0.5 to each connection. This is very controversial from the first intuition that connection weight decreases if the virtual nodes are further from each other. The authors need to prove if their method is reasonable.

Response:

Thank you very much for your comment. In this work, we have applied our physically implemented PBAONC complex network to reservoir computing. One of the most prominent advantages of reservoir computing is the simplicity of training that the reservoir itself is left untrained and only the readout layer is required to be trained. The underlying idea is that a randomly constructed reservoir offers a complex nonlinear dynamic transformation of the input signals which allows the readout to extract the desired output using a simple linear mapping. Although the exact weight distribution and sparsity is believed to have limited influence on the reservoir's performance, the best performing reservoirs have been shown to have spectral radii lower than one¹⁴. In constructing the theoretical models of reservoirs, the random weights are routinely drawn from a uniform distribution over $(-\epsilon, \epsilon)$ which are then rescaled to spectral radius less than unity^{15,16}.

As mentioned in our response to the reviewer's first comment, though weights are not designed intentionally in our approach, they are naturally present in our physically implemented complex network. Because each virtual node in our physically implemented PBAONC reservoir is connected to its subsequent ones within its resistance decay time with connection strengths decreasing with temporal separation, we have assigned distance dependent weights to these edges in the revised manuscript. Specifically, the weight is linearly decreased from 0.2 (connection to the immediately following node) as the connected node is farther away. For any node i , if $i + D_i \leq N$, the weight of the connection to its border node becomes zero; otherwise, the weight of the connection to node N is $\frac{0.2}{D_i} (D_i + i - N)$. As shown in Fig. 4c, the performance gap (as reflected by the proximity to unity) between PBAONC complex network and fully connected network under this more physically realistic weight assignment scheme is even larger, implying that memristive reservoirs have much room for improvement through the generation of complex networks.

This new weight assignment scheme and the corresponding new results have been described in the revised manuscript on page 13.

Comment #3:

The authors do not describe clearly what are the transient dynamical responses of the memristor that are used as input to the following linearly weighted matrix. Furthermore, the authors do not show how the transient dynamical responses are correlated to the memristor inputs. There need to be some experiment demonstrations showing the inputs and outputs of the generated network.

Response:

Thank you very much for your comment. In our work, we have used a single dynamical memristor with intrinsic variability repeatedly in a time-division multiplexed manner and generated a time-domain complex network composed of a

number of virtual nodes with internode couplings. This physically implemented complex network was further employed as a reservoir for reservoir computing, in which different temporal sequences of voltage pulses as inputs give rise to different trajectories of current evolutions. The reservoir state is represented by the instantaneous currents obtained when each of the N virtual nodes appears (I - if this virtual node is excited by a voltage pulse). These current values are then linearly weighted through an output weight matrix W_{out} and summed together to obtain the output of the reservoir computing system.

To illustrate how different temporal sequences of voltage pulses as inputs give rise to different trajectories of current evolutions, in the revised manuscript we have used pulse sequences of different intervals as inputs to drive the memristive reservoir to different states. As shown in Figs. S3b,c, when the multiplexing time slot θ is chosen to be 200 ms, which is smaller than the minimum resistance decay time τ_{min} of the dynamical memristor, a (001000101010111010110111) pulse train and a (111100011011000101101100) pulse train give rise to different current evolution trajectories, which forms the basis of input classification.

The operation of our physical reservoir has been further detailed in the revised manuscript on page 14.

Comment #4:

The authors do not clearly describe how and when the learning and testing processes are done. It would be great if workflow schematics can be used to describe the details, such as when the training data is input, when the weight matrix is updated, when the training is done and when the testing is performed.

Response:

Thank you very much for your comment. In the revised manuscript, we have also provided a workflow schematic to describe more clearly the procedures of learning and testing, as shown in Fig. S11.

Note that a major advantage of reservoir computing is fast training because only weights in the linear readout layer need to be trained, while the connection weights (not intentionally pre-designed but naturally present in our physical reservoir) in the reservoir remain fixed. In our experimental protocol, the weighted summation of the reservoir outputs and the final classification in the testing process, as well as the update of the weight matrix of the output layer are all performed on software.

Comment #5:

According to the authors' description, the transient dynamic responses of all the virtual nodes need to be recorded and to be further aligned. This could be very power consuming and time consuming. Besides, each memristor needs to be optimized separately to work with their optimum time slot, which is not promising for large-scale

applications and is also very inefficient. These are some intrinsic and crucial drawbacks of the approach in the manuscript.

Response:

Thank you very much for your comment. As the reviewer has pointed out, the serial feeding procedure (i.e., time division multiplexing) seems to result in a slow-down of the information processing compared to the parallel feeding procedure⁷. However, this potential slow-down is compensated for by considerable practical advantages that the reduction of a complex network to a single hardware node facilitates implementations enormously, because only a few components are needed, which may in turn reduce energy consumption significantly. In addition, the read-out can also be taken at a single point of the delay line. These simplifications will enable ultra-high-speed implementations, using high-speed components that would be too demanding or expensive to be used for many nodes⁷⁻⁹. As for our dynamical memristor as such a single physical node, it is a passive element with working current of only a few tens of nA and its speed limit could potentially be in the picosecond range¹⁰, thereby promising speed and energy advantages.

We have added these discussions in the revised manuscript on page 8.

As for your comment on the necessity of optimizing the time slot of each sequentially reused memristor, our observations (Figs. 5e,f) indicate that the most significant performance improvement results from the increase in the number of memristors, each functioning as a component reservoir. This improvement can be understood as due to device-to-device (D2D) variation. In addition, the use of a complex network as the reservoir also enhances the performance dramatically, which is enabled by the cycle-to-cycle (C2C) variation of the memristor. Based on the calculated memory capacity (Figs. 5a-d), we have constructed mixed reservoir sets by using many reservoirs with $D_{max} \in \{6, 7, 8, 9\}$ that have the largest memory capacity and supplementing with other reservoirs with $D_{max} \in \{3, 4, 5, 10, 11, 12, 13, 14\}$. These mixed reservoir sets may benefit from the richness of temporal dynamics. Though these mixed reservoir sets by design do have better performance in terms of memory capacity, respectable performance can already be achieved by simply increasing the number of component reservoirs (still much less hardware overhead compared to that of the conventional parallel feeding procedure) and engineering complex network topology into each individual reservoir (keeping $\theta \leq \tau_{min}$ and $N \times \theta \geq \tau_{max}$).

We have added these discussions in the revised manuscript on page 15.

Reviewer #3

General comments:

The manuscript presents an experimental implementation of a reservoir computer based on a single memristor and a time-multiplexing technique. Due to the time-multiplexing, an equivalent virtual network has a feed-forward structure, where the coupling between the virtual nodes occurs due to the finite current decay time in the device.

The main idea of this work is based on the intrinsic cycle-to-cycle variability. As a result of such a variability, the spontaneous current decay time is not constant and has a stochastic nature. Therefore, the resulting virtual network becomes non-regular. In addition, the virtual network can be controlled by varying the time-multiplexing stepsize. For sufficiently small stepsize, all virtual nodes are coupled, resulting in a Fully Coupled (FC) network. The time-multiplexing stepsize is the metaparameter that is used in the paper to switch between FC, small-world, or uncoupled networks.

Apart from the experimental implementation, the authors compare the performance of the obtained reservoir with the extreme FC case. This comparison shows that the performance increases significantly when the network topology becomes non-trivial (non-FC).

I find the idea of exploiting the intrinsic variability of the experimental device for reservoir computation very interesting. The intrinsic variability increases the complexity of the system and can potentially lead to improvement.

On the other hand, in my opinion, the paper does not show that the intrinsic variability is the source of the improvement. Perhaps I have misunderstood, but their only message regarding the performance is that their setup in the properly tuned regime with non-trivial network topology can perform better than the same setup in the FC regime. Whether the improvement is due to network variability or the fact that it is not FC is unclear.

Some other comments are listed below.

In my opinion, the paper is not suitable for publication in Nature Communications in the present form. A major revision may make it suitable, depending on the ability of the authors to refute the criticism.

Response:

Thank you very much for finding the idea behind this work interesting and giving us insightful comments to further improve the quality of this work. According to your comments, more experiments have been conducted to make the performance improvement as due to nontrivial network topology (thanks to device variability) unambiguous.

In addition, we conducted a more thorough review to more clearly illustrate the links and differences between the previous work and our work.

Our responses to your specific comments one by one are shown as follows.

Comment #1:

With reference to my comment above, can you show that the improvement in reservoir performance is related to the network variability?

Response:

Thank you very much for your question that has reminded us of a potential ambiguity in our previous manuscript about whether the improvement of reservoir performance is related to the nontrivial network topology arising from device variability or simply related to the sparse but regular network topology by regulating the multiplexing time slot θ . Because variability that underpins the generation of nontrivial network topology is an intrinsic property of our dynamical memristor, it is practically impossible to create a regular network through this approach. Therefore, we have carried out simulation study of the influence of network sparsity with and without randomness.

As introduced in the manuscript, a desired reservoir should exhibit a fading memory, that is, the effect of the previous reservoir state on a future state should vanish gradually as time passes¹⁷. Practically, this property is assured if the reservoir weight matrix W is scaled so that its spectral radius $\rho(W)$ (i.e., the largest absolute eigenvalue) satisfies $\rho(W) < 1$ ¹⁴. Theoretical analyses have also shown that a reservoir has an optimal active state if the $\rho(W)$ is close to 1¹⁶. As illustrated in Fig. 4c, the trivial AONC regular networks ($D_{min}=D_{max}=8$ or 2) without randomness in their connectivity patterns have $\rho(W)$ s that are less proximal to unity compared to that of the PBAONC complex network, though not as significant as the contrast between the PBAONC complex network and the PBAONC FC network.

We have added these results and discussions in the revised manuscript on page 13.

Comment #2:

Although time-multiplexing and time-folded virtual network is one of the main points used in the manuscript, previous main results on this topic are not properly introduced.

Response:

Thank you very much for your comment. On pages 4 and 12 of the revised manuscript, we have carried out more thorough literature survey and introduced the previous main results mainly from the viewpoint of network generation and reservoir computing application.

Comment #3:

The manuscript "promises" a scheme with on-demand generation of the network. However, the presented scheme is essentially based on the variation of the time-

multiplexing stepsize "theta" to achieve different coupling topologies. I am afraid that this possibility has already been explained in several previous studies. The authors did not mention this fact in the introduction.

Response:

Thank you very much for your comment. As the reviewer has pointed out, our proposed scheme is based on time multiplexing of a single dynamical memristor. By varying the time slot θ , networks in the time domain with different topologies, connection sparsity and coupling weights between nodes can be created. The proposal of using a single dynamical node with delayed feedback as a complex system was originally made by Appeltant et al.⁷ and has since been widely implemented in electronic and photonic devices¹⁸.

As for memristive implementations, Du et al.¹⁹ have used different time-multiplexing time slots for creating different component reservoirs. The motivation was to enrich the reservoir dynamics and benefit from device-to-device variation. Zhong et al.²⁰ have used a fixed total number of virtual nodes and a fixed time-multiplexing time slot, and investigated the optimal trade-off between the number of component reservoirs and the number of virtual nodes per reservoir. The coupling strength has effectively been tailored in these two cases. A more general framework of network emulation based on a single dynamical system with time-delayed feedback has recently been discussed by several groups^{13,21,22}. Among them, Stelzer et al.^{13,22} proposed the use of multiple delay loops with different delay lengths for constructing a deep neural network whose interlayer connection topology can be adjusted by the number of delay loops and the delay length of each loop (with a fixed multiplexing time slot and total number of virtual nodes). A key contribution of our work is improving the framework with regard to the emulation of complex networks with nontrivial topologies, as demonstrated by tuning the time slot of multiplexing a dynamical memristor with intrinsic variability. To the best of our knowledge, this is the first attempt to extend the framework in this direction and the first demonstrated hardware implementation as well.

We have added these introductions in the revised manuscript on pages 4 and 12.

Comment #4:

It is unclear where the ring topology for the visual representations of the virtual network in Figures 3,4 comes from. Is the last node N connected to the first node 1 via a delayed feedback? At least I did not find this in the description.

Response:

Thank you very much for your question. We have drawn the schematics in Figs. 3 and 4 following the seminal work on complex networks by Watts and Strogatz⁶. A ring over N nodes with edges individually represented by an arch arrow extending from one node to another offers visual simplicity.

For the sake of simplicity, the connection (if present) extended from the first node (at the clockwise end of the open ring) to the last node (i.e., the N th one counted in the clockwise direction) is represented by a short counterclockwise arrow covering the gap between them.

We have added this description in the caption of Fig. 3 in the revised manuscript.

Comment #5:

When the spectral radius of the reservoir is computed, random weights from the interval $[-0.5, +0.5]$ are assigned. It is unclear to me how these random weights relate to the experimental reservoir used for the computations. Are the weights between the virtual nodes additionally weighted? (I guess not). If these links are not weights, why are additional weights used for the spectral radius (especially negative values)?

Response:

Thank you very much for your questions. Our approach can create nontrivial network topologies as desired via controlling the multiplexing time slot θ , thanks to the variability of the resistance decay time τ (obeying a certain distribution) of the dynamical memristor. This approach is premised on the basis that the edge (unweighted) between any two virtual nodes is formed if they are dynamically coupled, or in other words, their temporal separation is less than τ . On the other hand, edges in a network are often associated with weights that differentiate them in terms of their strength. What we want to clarify here is that though weights are not designed intentionally in our approach, they are naturally present in our physically implemented complex network. Specifically, the connection strength between any two virtual nodes that are temporally separated by $m \times \theta$ can be reflected in the amplitude of the remanent current as the result of spontaneous decay over the period of $m \times \theta$ from I_+ excited at the moment when the former node appears (no further voltage excitation over this period). Accordingly, pairs of virtual nodes with different temporal separations will have different connection strength.

In this work, we have also applied our physically implemented PBAONC complex network to reservoir computing. One of the most prominent advantages of reservoir computing is the simplicity of training that the reservoir itself is left untrained and only the readout layer is required to be trained. The underlying idea is that a randomly constructed reservoir offers a complex nonlinear dynamic transformation of the input signals which allows the readout to extract the desired output using a simple linear mapping. Although the exact weight distribution and sparsity is believed to have limited influence on the reservoir's performance, the best performing reservoirs have been shown to have spectral radii lower than one¹⁴. In constructing the theoretical models of reservoirs, the random weights are routinely drawn from a uniform distribution over $(-\epsilon, \epsilon)$ which are then rescaled to spectral radius less than unity^{15,16}.

As mentioned above, though weights are not designed intentionally in our approach, they are naturally present in our physically implemented complex network.

Because each virtual node in our physically implemented PBAONC reservoir is connected to its subsequent ones within its resistance decay time with connection strengths decreasing with temporal separation, we have assigned distance dependent weights to these edges in the revised manuscript. Specifically, the weight is linearly decreased from 0.2 (connection to the immediately following node) as the connected node is farther away. For any node i , if $i+D_i \leq N$, the weight of the connection to its border node becomes zero; otherwise, the weight of the connection to node N is $\frac{0.2}{D_i}(D_i + i - N)$. As shown in Fig. 4c, the performance gap (as reflected by the proximity to unity) between PBAONC complex network and fully connected network under this more physically realistic weight assignment scheme is even larger, implying that memristive reservoirs have much room for improvement through the generation of complex networks.

This new weight assignment scheme and the corresponding new results have been described in the revised manuscript on page 13.

Comment #6:

The authors should clearly state that their virtual network is feed-forward, i.e., it is a directed network, where the earlier in time virtual nodes influence the later virtual nodes. In the literature on complex networks, FC networks are mostly considered to be bidirectional. Perhaps the directionality can also be shown in the figures. What is the coupling weight matrix for the typical (or best) choice of parameters?

Response:

Thank you very much for your suggestion. On page 11 of the revised manuscript, the unidirectional (feed-forward) property of our PBAONC networks implemented in the dynamical memristor has been clearly stated. Accordingly, the directionality is also shown in Fig. 3.

As for your question about the coupling weight matrix, we would like to stress again that though weights are not designed intentionally in our approach, they are naturally present in our physically implemented complex network. Specifically, the connection strength between any two virtual nodes that are temporally separated by $m \times \theta$ can be reflected in the amplitude of the remanent current as the result of spontaneous decay over the period of $m \times \theta$ from I_+ excited at the moment when the former node appears (no further voltage excitation over this period). Accordingly, pairs of virtual nodes with different temporal separations will have different connection strength.

In reservoir computing applications, the exact weight distribution and sparsity is believed to have limited influence on the reservoir's performance. Nevertheless, the best performing reservoirs have been shown to have spectral radii lower than one¹⁴. Because each virtual node in our physically implemented PBAONC reservoir is connected to its subsequent ones within its resistance decay time with connection

strengths decreasing with temporal separation, we have assigned distance dependent weights to these edges in the revised manuscript. Specifically, the weight is linearly decreased from 0.2 (connection to the immediately following node) as the connected node is farther away. For any node i , if $i+D_i \leq N$, the weight of the connection to its border node becomes zero; otherwise, the weight of the connection to node N is $\frac{0.2}{D_i}(D_i + i - N)$. It is seen from the short-term memory (STM) task (Fig. 5a) and parity check (PC) task (Fig. 5b) that large memory capacities are mainly achieved around $D_{max}=8$ and $N=20\sim 30$, where $\rho(W)$ s closest to 1 are achieved according to our weight assignment scheme (Fig. 4b).

This more physically realistic weight assignment scheme has been introduced in the revised manuscript on page 13.

References

- 1 Wu, W. *et al.* Improving analog switching in HfO_x-based resistive memory with a thermal enhanced layer. *IEEE Electron Device Letters* **38**, 1019-1022 (2017).
- 2 Kumar, S., Wang, X., Strachan, J. P., Yang, Y. & Lu, W. D. Dynamical memristors for higher-complexity neuromorphic computing. *Nature Reviews Materials*, 1-17 (2022).
- 3 Dalgaty, T. *et al.* In situ learning using intrinsic memristor variability via Markov chain Monte Carlo sampling. *Nature Electronics* **4**, 151-161 (2021).
- 4 Cai, F. *et al.* Power-efficient combinatorial optimization using intrinsic noise in memristor Hopfield neural networks. *Nature Electronics* **3**, 409-418 (2020).
- 5 Mahmoodi, M., Prezioso, M. & Strukov, D. Versatile stochastic dot product circuits based on nonvolatile memories for high performance neurocomputing and neurooptimization. *Nature communications* **10**, 1-10 (2019).
- 6 Watts, D. J. & Strogatz, S. H. Collective dynamics of ‘small-world’ networks. *nature* **393**, 440-442 (1998).
- 7 Appeltant, L. *et al.* Information processing using a single dynamical node as complex system. *Nature communications* **2**, 1-6 (2011).
- 8 Brunner, D., Soriano, M. C., Mirasso, C. R. & Fischer, I. Parallel photonic information processing at gigabyte per second data rates using transient states. *Nature communications* **4**, 1364 (2013).
- 9 Larger, L. *et al.* High-speed photonic reservoir computing using a time-delay-based architecture: Million words per second classification. *Physical Review X* **7**, 011015 (2017).
- 10 Menzel, S., Von Witzleben, M., Havel, V. & Böttger, U. The ultimate switching speed limit of redox-based resistive switching devices. *Faraday discussions* **213**, 197-213 (2019).
- 11 Milano, G. *et al.* In materia reservoir computing with a fully memristive architecture based on self-organizing nanowire networks. *Nature materials* **21**, 195-202 (2022).
- 12 Zhong, Y. *et al.* A memristor-based analogue reservoir computing system for real-time and power-efficient signal processing. *Nature Electronics* **5**, 672-681 (2022).
- 13 Stelzer, F., Röhm, A., Vicente, R., Fischer, I. & Yanchuk, S. Deep neural networks using a single neuron: folded-in-time architecture using feedback-modulated delay loops. *Nature communications* **12**, 1-10 (2021).
- 14 Jaeger, H. The “echo state” approach to analysing and training recurrent neural networks-with an erratum note. *Bonn, Germany: German National Research Center for Information Technology GMD Technical Report* **148**, 13 (2001).
- 15 Jaeger, H. & Haas, H. Harnessing nonlinearity: Predicting chaotic systems and saving energy in wireless communication. *science* **304**, 78-80 (2004).
- 16 Kawai, Y., Park, J. & Asada, M. A small-world topology enhances the echo state property and signal propagation in reservoir computing. *Neural Networks* **112**,

- 15-23 (2019).
- 17 Lukoševičius, M. & Jaeger, H. Reservoir computing approaches to recurrent neural network training. *Computer Science Review* **3**, 127-149 (2009).
- 18 Tanaka, G. *et al.* Recent advances in physical reservoir computing: A review. *Neural Networks* **115**, 100-123 (2019).
- 19 Du, C. *et al.* Reservoir computing using dynamic memristors for temporal information processing. *Nature communications* **8**, 1-10 (2017).
- 20 Zhong, Y. *et al.* Dynamic memristor-based reservoir computing for high-efficiency temporal signal processing. *Nature communications* **12**, 408 (2021).
- 21 Hart, J. D., Schmadel, D. C., Murphy, T. E. & Roy, R. Experiments with arbitrary networks in time-multiplexed delay systems. *Chaos: an interdisciplinary journal of nonlinear science* **27** (2017).
- 22 Stelzer, F. & Yanchuk, S. Emulating complex networks with a single delay differential equation. *The European Physical Journal Special Topics* **230**, 2865-2874 (2021).

REVIEWERS' COMMENTS

Reviewer #1 (Remarks to the Author):

The authors have addressed my comments reasonably well. I have no additional concerns.

Suhas Kumar

Reviewer #2 (Remarks to the Author):

I read the revised manuscript and responses carefully. The authors have addressed all the questions. Thus, I would recommend the publication of the manuscript.

Point-by-point response to reviewers' comments

Reviewer #1

General comments:

The authors have addressed my comments reasonably well. I have no additional concerns.

Suhas Kumar

Response:

Thank you very much for recommending the publication of our manuscript, and again, for spending your valuable time on reviewing the manuscript.

Reviewer #2

General comments:

I read the revised manuscript and responses carefully. The authors have addressed all the questions. Thus, I would recommend the publication of the manuscript.

Response:

Thank you very much for recommending the publication of our manuscript, and again, for spending your valuable time on reviewing the manuscript.